

# Remote sensing of volcanic $CO_2$, HF, HCl, $SO_2$, and BrO in the downwind plume of Mt. Etna

André Butz[1], Anna Solvejg Dinger[2], Nicole Bobrowski[2,3], Julian Kostinek[1], Lukas Fieber[2], Constanze Fischerkeller[1], Giovanni Bruno Giuffrida[4], Frank Hase[1], Friedrich Klappenbach[1], Jonas Kuhn[2], Peter Lübcke[2], Lukas Tirpitz[2], and Qiansi Tu[1]

[1]IMK-ASF, Karlsruhe Institute of Technology (KIT), Leopoldshafen, Germany
[2]Institute for Environmental Physics, Heidelberg University, Germany
[3]Institute of Geosciences, Johannes Gutenberg University Mainz, Germany
[4]Istituto Nazionale di Geofisica e Vulcanologia, Palermo, Italy

*Correspondence to:* André Butz (andre.butz@gmail.com)

**Abstract.** Remote sensing of the gaseous composition of non-eruptive, passively degassing volcanic plumes can be a tool to gain insight into volcano interior processes. Here, we report on a field study in September 2015 that demonstrates the feasibility of remotely measuring the volcanic enhancements of carbon dioxide ($CO_2$), hydrogen fluoride (HF), hydrogen chloride (HCl), sulfur dioxide ($SO_2$), and bromine monoxide (BrO) in the downwind plume of Mt. Etna using portable and rugged

spectroscopic instrumentation. To this end, we operated the Fourier Transform Spectrometer EM27/SUN for the shortwave-infrared (SWIR) spectral range together with a co-mounted UV spectrometer on a mobile platform in direct-sun view at 5 to 10 km distance from the summit craters. The three days reported here cover several plume traverses and a sunrise measurement. For all days, intra-plume HF, HCl, $SO_2$, and BrO vertical column densities (VCDs) were reliably measured exceeding $5\times10^{16}$ molec/cm$^2$, $2\times10^{17}$ molec/cm$^2$, $5\times10^{17}$ molec/cm$^2$, and $1\times10^{14}$ molec/cm$^2$, with an estimated precision of

$2.2\times10^{15}$ molec/cm$^2$, $1.3\times10^{16}$ molec/cm$^2$, $3.6\times10^{16}$ molec/cm$^2$, and $1.3\times10^{13}$ molec/cm$^2$, respectively. Given that $CO_2$, unlike the other measured gases, has a large and well-mixed atmospheric background, derivation of volcanic $CO_2$ VCD enhancements ($\Delta CO_2$) required compensating for changes in altitude of the observing platform and for background concentration variability. The first challenge was met by simultaneously measuring the overhead oxygen ($O_2$) columns and assuming covariation of $O_2$ and $CO_2$ with altitude. The atmospheric $CO_2$ background was found by identifying background soundings

via the co-emitted volcanic gases. The inferred $\Delta CO_2$ occasionally exceeded $2\times10^{19}$ molec/cm$^2$ with an estimated precision of $3.7\times10^{18}$ molec/cm$^2$ given typical atmospheric background VCDs of 7 to $8\times10^{21}$ molec/cm$^2$. While the correlations of $\Delta CO_2$ with the other measured volcanic gases confirm the detection of volcanic $CO_2$ enhancements, correlations were found of variable significance ($R^2$ ranging between 0.88 and 0.00). The intra-plume VCD ratios $\Delta CO_2/SO_2$, $SO_2$/HF, $SO_2$/HCl, and $SO_2$/BrO were in the range 7.1 to 35.2, 5.02 to 10.5, 1.54 to 3.43, and $2.9\times10^3$ to $12.5\times10^3$, respectively, showing pronounced

20 day-to-day and intra-day variability.





## 1 Introduction

The gaseous composition of non-eruptive, passively degassing volcanic plumes is largely dominated by water vapor ($H_2O$), carbon dioxide ($CO_2$), sulfur dioxide ($SO_2$), hydrogen sulfide ($H_2S$), and halogen bearing compounds such as hydrogen fluoride (HF), hydrogen chloride (HCl), and bromine monoxide (BrO). The relative abundances of the constituents vary with the type of volcano and magmatic composition, the contributing degassing mechanisms, and the dynamics of mass transport in the volcano interior. Thus, monitoring volcanic gas emissions can help constrain subsurface processes and estimate fluxes of the geological carbon cycle (e.g. Allard et al., 1991; Goff et al., 2001; Burton et al., 2013). Furthermore, it holds the promise for improved eruption forecast since enhanced $CO_2/SO_2$ emission ratios have been shown to precede eruptive volcanic activity with a lead time of hours to weeks (e.g. Aiuppa et al., 2007, 2010).

While remote sensing of volcanic plume composition has been demonstrated a valuable tool for various gases including HF, HCl, $SO_2$, and BrO (e.g. Mori et al., 1993; Love et al., 1998; Francis et al., 1998; Bobrowski et al., 2003; Grutter et al., 2008; Stremme et al., 2011, 2012; Theys et al., 2013), measuring volcanic $CO_2$ remotely faces the particular challenge to discriminate the volcanic enhancements from the background concentration of about 400 ppm (parts per million per volume). Therefore, measurement techniques for volcanic $CO_2$ mostly rely on *in-situ* gas analyzers (e.g. Shinohara et al., 2008), open-path or closed-path spectroscopic techniques (e.g. Burton et al., 2000; Gerlach et al., 2002) deployed in the proximity of the source region, thereby avoiding downwind dilution of the plume. However, deployment close to the source typically comes with great costs and logistics effort, various hazards for instruments and operators and, depending on the employed sampling strategy, limited representativeness for the volcanic source as a whole. Pioneering remote sensing experiments relied on detecting the absorption of volcanically emitted $CO_2$ along an atmospheric light path using the infrared emission of hot volcanic material (Naughton et al., 1969; Mori and Notsu, 1997) or exploiting the thermal contrast between the hot volcanic plume and the background sky (Goff et al., 2001). Aiuppa et al. (2015) demonstrated a LIDAR technique for scanning a volcanic $CO_2$ plume. Yet, these remote sensing techniques require path-integrated $CO_2$ enhancements in excess of several ppm or thermal contrast of several degrees between the plume and the background. Thus, they need to sample the plume in the proximity of the source.

Here, we demonstrate that our field-deployable spectroscopic instrumentation enables remote sensing of volcanic $CO_2$ enhancements simultaneously with co-emitted HF, HCl, $SO_2$, and BrO. Our setup builds on direct-sunlight spectroscopy using portable spectrometers that sample the plume several kilometers downwind of Mt. Etna's craters. The key enabling element is the Fourier Transform Spectrometer (FTS) EM27/SUN that allows for measuring path-integrated $CO_2$ enhancements well below 1 ppm exploiting shortwave-infrared (SWIR) absorption bands (Gisi et al., 2012; Hase et al., 2015; Frey et al., 2015; Klappenbach et al., 2015). The FTS also allows for measuring path-integrated HF and HCl concentrations. We operated two of the FTS, one stationary and another one on a small truck together with a co-mounted grating spectrometer targeting $SO_2$ and BrO in the ultraviolet (UV) spectral range. The mobile FTS and the UV spectrometer were fed by the same beam of direct sunlight from a custom-built solar tracker (Klappenbach et al., 2015). During a 3-week field campaign in Sep. 2015, the truck carried the mobile remote sensing observatory to various sampling locations in the vicinity of Mt. Etna while the stationary FTS observed background variability for most of the time.



## 2 Campaign deployment at Mt. Etna

Mt. Etna is a quiescently degassing volcano on Sicily, Italy, exhibiting four active summit craters in a confined source region above 3,100 m altitude. Typical passive, continuous non-eruptive degassing amounts to more than 5 ktons of $CO_2$ per day (e.g. Aiuppa et al., 2008; Burton et al., 2013). Eruptive activity occurs occasionally but not during the campaign operations reported

here, starting on Sep. 5 and ending on Sep. 25, 2015. Whenever weather conditions were favorable for direct-sun view, the custom-built solar tracker, the mobile FTS, and the UV spectrometer were mounted on the truck. Figure 1 illustrates the setup of the mobile observatory. In the morning, the truck was taken to a location where visual inspection of the sky indicated Etna's plume roughly overhead. During sunrise, at large solar zenith angles, the observatory remained stationary collecting absorption spectra while the sun was rising. Later, at smaller solar zenith angles, we operated the observatory in stop-and-go patterns to

collect scans through the volcanic plume. Operations were suspended whenever the sun was obscured by cloud cover, which persisted for several days in early September due to rainy conditions. Generally, cloud cover appeared around local noon. The early phase of the campaign was dedicated to testing the setup and identifying suitable measurement procedures. In total, we report on 3 days (Sep. 18, Sep. 22, Sep. 23, 2015) in the second half of the campaign, for which we focus on stop-and-go patterns in the vicinity of Rifugio G. Sapienza (37.700°N, 14.999°W, 1,910 m a.s.l.) at the southern slope of Mt. Etna.

The typical distance between the volcanic source region at the summit and the positions of the sun-viewing spectrometers amounted to 5-10 km. Figure 2 shows an illustrative trajectory of the truck on Sep. 22, 2015. Since the truck was restricted to use general purpose roads, deployment options were limited and it was impossible to collect cross-sections that were strictly perpendicular to the plume direction. Figure 2 also demonstrates that, due to rough topography in the vicinity of Mt. Etna, lateral displacements of the truck imply changes in observer altitude. Note that the mobile spectrometers were deployed south

of the craters in such a way that the observation vectors towards the sun had southward components, sampling the plume at greater distances than the deployment locations. In addition to the mobile observatory, we operated another FTS at our base station in Milo (37.731°N, 15.113°W, 806 m a.s.l.) at the eastern slope of Mt. Etna. For the days reported here, the stationary FTS mostly observed background airmasses providing a reference for the background variability of $CO_2$.

## 3 Instrumentation

Both, the mobile and the stationary FTS are of the type EM27/SUN developed by Bruker Optics and the Karlsruhe Institute of Technology. The EM27/SUN delivers solar absorption spectra at a spectral resolution of $0.5\,cm^{-1}$ in the range 5,000 to $11,000\,cm^{-1}$ without the need for detector cooling. Gisi et al. (2012), Hase et al. (2015) and Frey et al. (2015) showed that stationary deployment of the EM27/SUN enables measuring the $CO_2$ column concentrations with high accuracy and precision. Klappenbach et al. (2015) detailed the performance of the mobile instrument used here, for deployment on a research vessel.

Previous studies typically co-added 10 double-sided interferograms with a total exposure of roughly 60 s. We adopted the same measurement pattern for the stationary FTS, but for the mobile FTS, we processed pairs of double-sided interferograms with a total exposure of 12 s in order to keep temporal averaging small. The total field-of-view of the FTS amounts to 0.27° resulting in the central part of the solar disk being observed.





The UV spectrometer is an Avantes ULS2048x64-ENV5 grating spectrometer, which operates in the spectral range 294 to 457 nm with a spectral resolution of 0.80 nm. The field-of-view is 1.15° and observes the whole solar disk with a factor 2 margin. We typically co-added several spectra resulting in total exposure times between a few hundred milliseconds and a few seconds depending on solar elevation. The UV spectrometer was kept at a constant temperature of 15° C.

While the stationary FTS was equipped with the standard solar tracker delivered by Bruker Optics (Gisi et al., 2011), the mobile observatory used a custom-built variant, initially developed for mobile applications by Klappenbach et al. (2015) and further enhanced here, to improve on the response times and to feed the mobile FTS and the UV spectrometer with the same light beam. Two rotatable tracking mirrors direct a light beam of 40 mm diameter into the mobile FTS through a wedged window. Given that the FTS only uses the central part of the beam, a portion of the outer part of the beam is fed into the UV

spectrometer through a telescope attached to a glass fiber bundle. Compared to previous applications, volcanic $CO_2$ plume detection benefited from enhancing the response time of the solar tracker. At moderate driving speed (up to ∼30 km/h) of the platform and benign road surface, the solar tracker was able to reliably stay on target (the sun) while driving and tracking stability was sufficient to enable recording of absorption spectra by the fast UV spectrometer. Real-time analysis of the $SO_2$ absorption signal on the passenger seat guided the stop-and-go patterns. However, tracking stability while driving was not suffi-

cient to record $CO_2$ absorption spectra by the mobile FTS, which has substantially longer exposures than the UV spectrometer. Thus, all $CO_2$ measurements discussed here were recorded while the truck was stopped.

## 4   Data analysis

The solar absorption spectra recorded by the FTS and the UV spectrometer provide information on the gas concentrations, integrated along the path from the sun to the ground-based observer. We first retrieved the column gas abundances of $CO_2$,

HF, and HCl from the SWIR (section 4.1) and the abundances of $SO_2$ and BrO from the UV spectra (section 4.2). Then, we inferred the volcanic enhancements (section 4.3).

### 4.1   Spectral retrieval in the SWIR

The FTS delivered interferograms, from which we generated absorption spectra through Fourier Transformation assuming Norton-Beer's medium strong apodization function (Griffiths and de Haseth, 2007). The DC-part of the interferogram is a good

measure for brightness fluctuations of the light source during recording of the interferogram. As described in Klappenbach et al. (2015), we used the DC-part to discard spectra which suffered from variable illumination e.g. due to unstable tracking of the sun. Small fluctuations of the DC-part that passed our quality filter were corrected as described in Keppel-Aleks et al. (2007). In addition to the interferograms with severe brightness fluctuations, we discarded FTS measurements in cases of generally inferior quality of the spectra (as judged on basis of fitting residuals) and unsuccessful convergence of the spectral retrieval.

Overall, the quality filtering removed virtually all the spectra recorded by the mobile FTS while the truck was moving.

After generation of the absorption spectra, a line-by-line atmospheric transmittance model retrieved the concentrations of the target gases integrated along the vertical, the vertical-column-densities (VCDs). We employed the software package "PROF-





FIT" developed for ground-based direct-sun spectroscopy (Hase et al., 2004) and previously used for processing EM27/SUN spectra (Gisi et al., 2012; Frey et al., 2015; Hase et al., 2015; Klappenbach et al., 2015). PROFFIT evaluates Beer-Lambert's law assuming a curved hydrostatic atmosphere with horizontally homogeneous layers. We considered several spectral windows in the SWIR with various absorbing molecules as detailed in table 1. Molecular absorption was taken into account

through a Voigt line-shape model including corrections due to line-mixing for $CO_2$. Table 1 also lists the databases for the driving spectroscopic parameters. An empirical linelist modeled the solar Fraunhofer lines. The parameters to be estimated for each measurement comprised scaling factors for the vertical *a priori* profiles of the absorbers and auxiliary parameters such as spectral shift parameters per window and parameters to fit the transmittance baseline. For $CO_2$, HCl, and HF, we only scaled the lower tropospheric part of the vertical profile and adopted the *a priori* for the upper tropospheric and stratospheric part.

VCDs are *a posteriori* calculated by summing up the layers of the scaled vertical profiles. The volcanic VCD enhancements discussed below thus assume the case of a horizontally homogeneous, extended plume.

Since the molecular absorption lines in the SWIR are optically thick as well as pressure and temperature dependent, the spectral retrieval in the SWIR (unlike DOAS in the UV) requires a detailed representation of the vertical profiles of meteorological parameters and gas concentrations. The profiles of pressure, temperature, and humidity were derived from global

NCEP (National Centers for Environmental Prediction) fields on $1° \times 1°$ (latitude$\times$longitude) via interpolation in space and time. Surface pressure, surface elevation, latitude and longitude of the soundings were measured by pressure transducers and GPS recorders deployed next to the FTS and subsequently used to construct the input meteorological profiles and the *a priori* absorber profiles. For $O_2$ and $H_2O$, the latter were taken from the meteorological pressure and humidity data. For $CO_2$, HCl, HF, and $CH_4$, we used climatological profiles. Note that all retrievals irrespective of sampling intra-plume or background con-

ditions assumed background absorber profiles i.e. the retrieval had no *a priori* information on the volcanic plume except for the fact that we only allowed for scaling of the lower tropospheric part of the vertical profile.

## 4.2 Spectral retrieval in the UV

The spectral retrieval for the soundings of the UV spectrometer was based on the DOAS (Differential Optical Absorption Spectroscopy) technique (Platt and Stutz, 2008). It builds on Beer-Lambert's law and exploits narrow-band absorption bands

of optically thin absorbers like $SO_2$ and BrO. Table 2 lists details of the spectral windows and the absorption cross sections used for the target gas retrieval. The tabulated high-spectral-resolution absorption cross sections were convolved by the instrument spectral response function and then used to fit the recorded transmittance spectra. A least-squares fitting routine determined the target absorber concentrations integrated along the slant lightpath through the atmosphere, the slant-column-densities (SCDs). Ancillary fitting parameters were a cubic background polynomial, a linear additive offset accounting for spectrometer straylight

as well as the SCDs of interfering absorbers such ozone ($O_3$), nitrogen dioxide ($NO_2$), the oxygen collisional complex ($O_2$-$O_2$), and formaldehyde ($CH_2O$). UV spectra were discarded in cases of unstable temperature of the spectrometer and strong intensity fluctuations between adjacent spectra.

To compare the UV-retrieved SCDs with the VCDs retrieved in the SWIR, we translated the $SO_2$ and BrO SCDs into VCDs. To this end, we approximated the geometric assumptions of the SWIR-retrieval by ratioing the SCDs by the airmass factor





$\frac{1}{cos(SZA)}$ (solar zenith angle SZA). If the volcanic plume was horizontally extended and if our lines-of-sight crossed the plume perpendicularly to its propagation direction, the reported VCDs were truly representative of the vertical column enhancement in the plume. If the volcanic plume has a more complicated geometric shape, that shape would need to be considered for inferring volcanic enhancements and volcanic emissions in absolute units. Ratios of the various gases as discussed below, however, are

not affected by the geometric conversion since all gases were measured in the same lightpath.

## 4.3 Deriving volcanic VCD enhancements

Whenever we detected significant VCDs of $SO_2$, HCl, HF, and BrO, we considered those to be volcanic enhancements since atmospheric background concentrations of these species are small. Atmospheric $CO_2$, however, is a generally well-mixed gas with background concentrations of roughly 400 ppm (VCD $\sim 8.5 \times 10^{21}$ molec/cm$^2$ at 1 bar surface pressure). Therefore,

determining volcanic $CO_2$ enhancements faces the challenges that A) the measured VCDs co-vary with observer altitude and B) the atmospheric background needs to be removed. We addressed challenge A) by simultaneously retrieving the VCD of molecular oxygen ($O_2$) and calculating the column-average dry-air mixing ratio XCO$_2$ via

$$XCO_2 = \frac{[CO_2]}{[O_2]} \times 0.20942, \tag{1}$$

where square brackets ([ ]) indicate observed VCDs and 0.20942 is the atmospheric $O_2$ mixing ratio. Referencing by retrieved $O_2$ is a standard method that aims at canceling instrumental effects and retrieval artefacts (e.g. Wunch et al., 2010;

Klappenbach et al., 2015). Assuming that a change in observer altitudes causes the same relative change in the $O_2$ and $CO_2$ VCDs, XCO$_2$ derived from equation (1) is independent of observer altitude and thus, serves to address challenge B).

The atmospheric XCO$_2$ background varies due to variable meteorological conditions transporting distant source/sink signals to Mt. Etna. Further, imperfect knowledge of spectroscopic parameters and other instrumental or retrieval effects can cause a spurious dependence of XCO$_2$ on viewing geometry (e.g. Klappenbach et al., 2015). Therefore, we fitted a function $P$ linear in

time to the background XCO$_2$ records. Background soundings are the ones collected at most 30 min before or after a plume scan and with small HF VCDs ([HF]$<1 \times 10^{16}$ molec/cm$^2$). Intra-plume measurements are those in-between background soundings exceeding the HF threshold. The volcanic column-average mixing ratio enhancement $\Delta$XCO$_2$ then reads

$$\Delta XCO_2 = XCO_2 - P. \tag{2}$$

The volcanic VCD enhancement $\Delta CO_2$ is given by

$$[\Delta CO_2] = \Delta XCO_2 \times \frac{[O_2]}{0.20942}. \tag{3}$$

Figures 3 through 5 illustrate the step-by-step derivation of volcanic $\Delta$XCO$_2$ for the three considered days. The variation

of the retrieved $CO_2$ VCDs (upper panels in figures 3 through 5) is mostly due to changes in observer altitude. On Sep. 18, the stop-and-go operations started at around 6 h UTC at high altitude close to Rifugio G. Sapienza corresponding to low $CO_2$ VCDs. The local roads then led the truck downhill toward the south-west causing increasing $CO_2$ VCDs until we returned to Rifugio G. Sapienza at the end of the morning at about 9 h UTC. On Sep. 22, stop-and-go patterns were restricted to the closer



vicinity of Rifugio G. Sapienza (see also figure 2) with less altitude-induced changes in $CO_2$ VCDs than on the other days. On Sep. 23, observations started around 6 h UTC in stationary configuration at Rifugio G. Sapienza waiting for the sun to rise. Around 6:45 h UTC, we took up the stop-and-go patterns which again led us down and back up Mt. Etna's southern slope until 9 h UTC.

Calculating $XCO_2$ according to equation (1) removes the altitude-induced variability in the observed $CO_2$ VCDs (middle panels in figures 3 through 5) and reveals distinct volcanic enhancements in-between the background measurements and for the sunrise observations on Sep. 23. Data scatter grows toward late morning which might be due to some spectra with non-perfect solar tracking escaping our quality filters. Typically, clouds developed in late morning and disrupted the solar tracking. The $XCO_2$ records observed by the stationary FTS deployed at Milo on the eastern slope of Mt. Etna (blue crosses) confirm that

the linear background polynomial (red) is well suited to account for background $XCO_2$ variability. Except for late morning on Sep. 23, the stationary FTS at Milo sampled background airmasses (according to the HF threshold). On Sep. 23 after about 8:40 UT, the volcanic plume started drifting toward the lines-of-sight of the stationary FTS. We refrained from directly using the stationary measurements for background removal since the stationary FTS was placed at about 10 km distance from the truck trajectory at the eastern slope of the mountain. An improved setup might benefit from operating the stationary FTS closer

to the trajectory of the mobile observatory.

Removing the background via equation (2) yields the volcanic enhancement $\Delta XCO_2$ (lower panels in figures 3 through 5) which is further processed to yield the volcanic VCD enhancement according to equation (3).

## 5   Volcanic $\Delta CO_2$, HF, HCl, $SO_2$, and BrO

Figures 6 through 8 collect time-series of the volcanic enhancements of $CO_2$, HF, HCl, $SO_2$, and BrO found in the plume of

20   Mt. Etna. Our mobile observatory detected the volcanic plume for a single stop-and-go traverse on Sep. 18, three traverses on Sep. 22, and another traverse as well as the sunrise measurements on Sep. 23. Since the truck carrying the observatory was constrained to use the local roads, the traverses were not strictly perpendicular to the plume but exhibited displacements along plume direction and in altitude (see also Figure 2). Early on Sep. 18, for example, Etna's plume was sampled while the observatory was moving to the south-west until 8 h UTC, when a dead end forced us to turn back and to move the truck to the

east.

We find volcanic $\Delta CO_2$ occasionally exceeding $2 \times 10^{19}$ molec/cm$^2$ which, depending on observer altitude, amounts to column-average mixing ratio enhancements of a few tenths of a ppm (see also lower panels in Figures 3 through 5). Thus, given background VCDs of 7 to $8 \times 10^{21}$ molec/cm$^2$, the detected volcanic signal corresponds to a sensitivity of about 1:400. We estimated the overall $\Delta CO_2$ precision from the standard deviation (1-$\sigma$) of all the background measurements identified

via the HF threshold. It amounts to $3.7 \times 10^{18}$ molec/cm$^2$ suggesting that individual $\Delta CO_2$ measurements exceed the precision estimate by a factor 5 to 6. Detection of $\Delta CO_2$ is corroborated by correlated enhancements in HF, HCl, $SO_2$, and BrO. Precision estimates for the latter species amount to $2.2 \times 10^{15}$ molec/cm$^2$, $1.3 \times 10^{16}$ molec/cm$^2$, $3.6 \times 10^{16}$ molec/cm$^2$, and $1.3 \times 10^{13}$ molec/cm$^2$, respectively, where we first averaged the UV-measured $SO_2$ and BrO VCDs on the 12 s integration time





of coincident FTS spectra and then, calculated the standard deviation of the background measurements identified through the HF criterion.

Figure 9 further examines the correlation between $\Delta CO_2$ and co-emitted $SO_2$ for the intra-plume measurements while Figures 10 through 12 show the correlations between $SO_2$ and HF, HCl, BrO. As above the UV-measured $SO_2$ and BrO VCDs

were averaged for the integration time of coincident FTS spectra. Table 3 summarizes the $\Delta CO_2/SO_2$, $SO_2/HF$, $SO_2/HCl$, and $SO_2/BrO$ VCD ratios (as well as $R^2$) obtained by fitting straight lines to the correlations. For Sep. 22 and 23, we investigated the correlations for all the intra-plume measurements of the respective day as well as for 3 subsets on each day. On Sep. 22, these subsets correspond to the 3 stop-and-go transects (before 8:00 h UTC, between 8:00 and 8:30 h UTC, after 8:30 h UTC). On Sep. 23, the correlations group into 3 subsets that can be attributed to the sunrise observations, the early stop-and-go operations

(between 7:00 and 7:45 h UTC), and the later stop-and-go operations (after 7:45 h UTC).

The $\Delta CO_2/SO_2$ ratios show considerable day-to-day and intra-day variability ranging between $7.1 \pm 1.5$ and $35.4 \pm 1.3$ when considering the intra-day subsets. The errors correspond to the standard deviations of the fitted slopes. On Sep. 18, $R^2$ documents a good correlation between $\Delta CO_2$ and $SO_2$. On Sep. 22, $R^2$ is fair for the 2 earlier plume transects but vanishes for the third transect and for the case of a single correlation assumed valid for the whole day. The time series in Figure 7 confirms

that low outliers contaminate the $\Delta CO_2$ record after 8:30 h UTC on Sep. 22, which might be due to emerging cloud cover disturbing solar tracking stability. On Sep. 23, we find good correlation for the sunrise observations and a fair $R^2$ for the early phase of the plume transect (between 7:00 and 7:45 h UTC). For the later phase of the transect (after 7:45 h UTC), $\Delta CO_2$ is negative which is unreasonable and possibly points to deficiencies of the $CO_2$ background removal as further discussed below. Assuming a single correlation for Sep. 23 would clearly mask the geophysical variability contained in the records.

The $SO_2/HF$, $SO_2/HCl$, and $SO_2/BrO$ VCD ratios reveal high $R^2$ for all intra-day subsets. On Sep. 23, there is considerable variability in the VCD ratios which is supported by Figures 10 through 12. The decreasing $SO_2/HF$, $SO_2/HCl$, and $SO_2/BrO$ ratios indicate that the composition of the sampled parts of the plume changed from relatively $SO_2$-rich to $SO_2$-poor in the course of the morning hours. The $\Delta CO_2/SO_2$ ratios also yield a substantial decrease between sunrise and the early stop-and-go operations (between 7:00 and 7:45 h UTC), but an increase later (after 7:45 h UTC). After 7:45 h UTC, however, $\Delta CO_2$ is

negative which is unreasonable. So, we argue that, on Sep. 23, composition of the observed parts of the plume changed from relatively $SO_2$-rich to $SO_2$-poor, and likewise, from relatively $CO_2$-rich to $CO_2$-poor while HF, HCl, and BrO levels remained relatively elevated. The argument is supported by the observations of the stationary FTS which started detecting elevated HF levels at the eastern slope from 8:40 h UTC onward. This finding suggests that, in the later phase of the stop-and-go operations on the southern slope, the plume migrated away from the lines-of-sight of the mobile observatory toward the lines-of-sight of

the stationary FTS deployed on the eastern slope. Most likely, this change went along with variable contributions of the various craters to the plume composite observed by the mobile FTS.

Given the detected composition variability for Sep. 23, the negative $\Delta CO_2$ found after 7:45 h UTC could be an artefact of the $CO_2$ background removal which is based on the HF threshold for identifying background soundings. If our spectrometers sampled a $CO_2$-rich (and HF-rich) plume in the early morning and a $CO_2$-poor (but HF-rich) plume later, HF would be a poor

$CO_2$-plume indicator for the later period and the negative $\Delta CO_2$ could just correspond to undetectably low enhancements. To





try the sensitivity of the inferred $\Delta CO_2$ to background removal on Sep. 23, we deliberately categorized all $CO_2$ measurements after 7:45 h UTC as background and recalculated $\Delta CO_2$ as described in section 4.3. The inferred $\Delta CO_2/SO_2$ ratios for sunrise and for the early stop-and-go operations (between 7:00 and 7:45 h UTC) remain largely unchanged, amounting to 34.8±0.9 ($R^2$=0.88) and 12.7±2.7 ($R^2$=0.21), respectively. Thus, our conclusions for the early measurements on Sep. 23 are largely

insensitive to background removal. Overall, our assessment emphasizes that short-term composition variability of the observed plume requires careful consideration.

## 6  Discussion and conclusion

We demonstrate simultaneous remote sensing of volcanic $CO_2$ ($\Delta CO_2$), HF, HCl, $SO_2$, and BrO VCD enhancements in the plume of quiescently degassing Mt. Etna several kilometers downwind of the source. Our remote sensing observatory combined

the portable and rugged EM27/SUN FTS for the SWIR spectral range (observing $CO_2$, HF, HCl) with a DOAS spectrometer for the UV (observing $SO_2$, BrO), both instruments measuring direct-sun absorption spectra. The spectrometers were supplied with sunlight by a common, fast solar tracker, all together deployed on a mobile platform and supplied by a 12 Volts battery. The mobile setup enabled sequentially measuring intra-plume and background airmasses in stop-and-go patterns. Campaign operations were supported by another, stationary EM27/SUN FTS, sampling atmospheric background conditions for most of

the time. Generally, we focused on the retrieval of $\Delta CO_2$, since $CO_2$, unlike the other measured volcanic gases, has a high atmospheric background concentration that is well-mixed.

For three days reported here, the sunrise and stop-and-go observations yielded $\Delta CO_2$ up to about $2 \times 10^{19}$ molec/cm$^2$ with an estimated precision of $3.7 \times 10^{18}$ molec/cm$^2$. The other volcanic gases were measured with an estimated precision of $2.2 \times 10^{15}$ molec/cm$^2$, $1.3 \times 10^{16}$ molec/cm$^2$, $3.6 \times 10^{16}$ molec/cm$^2$, and $1.3 \times 10^{13}$ molec/cm$^2$, for HF, HCl, $SO_2$, and BrO, respectively.

Key to detecting the small volcanic $\Delta CO_2$ enhancements on top of the high atmospheric $CO_2$ background column (7 to $8 \times 10^{21}$ molec/cm$^2$) was the simultaneous observation of the overhead $O_2$ column and of volcanic HF co-emitted with $CO_2$. The $O_2$ column was used to compensate $CO_2$ variations due to changes in observer altitude. The HF columns provided an indication for intra-plume and background measurements. The latter were used to remove the atmospheric $CO_2$ background. The tightest correlation between $\Delta CO_2$ and $SO_2$ ($R^2$=0.88) was found for the sunrise observations on Sep. 23 where the slant absorption

path through the atmosphere was longest among our campaign records. Good to fair correlations ($R^2$ in the range 0.45 to 0.17) were found for 4 stop-and-go plume transects. One plume transect yielded a vanishing correlation between $\Delta CO_2$ and $SO_2$ and another plume transect yielded negative $\Delta CO_2$, most likely related to a change of plume composition from $CO_2$-rich to $CO_2$-poor during the transect. The correlations between $SO_2$ and HF, $SO_2$ and HCl, and $SO_2$ and BrO were all high ($R^2$ greater than 0.9 for all except one transect).

The intra-plume VCD ratios of $\Delta CO_2/SO_2$, $SO_2/HF$, $SO_2/HCl$, and $SO_2/BrO$ match the range of emission ratios and molar ratios previously reported for Mt. Etna (e.g. Aiuppa et al., 2007, 2008; La Spina et al., 2010; Voigt et al., 2014). During our campaign operations, *in-situ* gas analyzers deployed in the vicinity of the summit craters found molar $CO_2/SO_2$ ratios of 17.7±10.3 close to Voragine crater on Sep. 18, 22.2±5.8 close to Bocca Nuova crater on Sep. 22, and 13±5 close to Bocca





Nuova crater on Sep. 23, which confirms that Mt. Etna exhibits substantial variability of the emissions and that the various craters show different emission ratios (e.g. Aiuppa et al., 2008; La Spina et al., 2010). Since our remote sensing approach samples the plume several km downwind of the craters, variability of the observed plume composition can be either due to temporally variable source gas emission, varying contributions of the different craters (e.g. induced by variable winds), and

5 possibly flank emissions contributions (Allard et al., 1991) to the plume. Interpretation of our remote sensing data records becomes difficult if such variability occurs on the time scales of a plume transect. For our measurements on Sep. 23, we find a change in plume composition from $CO_2$-rich and $SO_2$-rich to $CO_2$-poor and $SO_2$-poor in the coarse of the morning hours. Most likely, the detected trend occurred since plume direction changed from southward to eastward which caused variable contributions of the various craters to the plume composite observed on the southern slope of Mt. Etna.

The employed methodology bares potential for substantial extension and refinement. In the view of plume variability, data interpretation would benefit from reducing the time span needed to conduct plume transects by the mobile observatory. During our campaign, the fastest transects took roughly 30 min. Reducing the exposure time of the FTS or the number of spectra collected during a transect would speed-up the operations but signal-to-noise would become worse. A caveat also applies to variability of the volcanic $CO_2$, HF, or HCl signal on the time scale of the FTS exposure (12 s for the mobile FTS, 60 s for

the stationary FTS). Since the FTS collects interferograms, a change of the volcanic gas signal during interferogram recording has a largely unpredictable effect on the absorption spectra and the inferred gas columns. A grating spectrometer would in good approximation average the volcanic signal over the exposure time. In the view of extending our methodology toward emission estimates, we will aim at linking the VCD ratios observed in the plume to actual emission ratios by combining our measurements with the monitoring infrastructures at Mt. Etna such as the FLAME $SO_2$ network (Salerno et al., 2009), by

operating a dedicated $SO_2$ camera system together with the direct-sun spectrometers (e.g. Kern et al., 2015; D'Aleo et al., 2016), or by meteorological modelling of the local wind fields. In the view of volcano monitoring, a few of our spectrometers could be operated in stationary observatories, possibly together with a mobile observatory, to setup a network that allows for monitoring Mt. Etna's or any other suitable volcano's plume. If the FTS can be deployed somewhat closer to the source than the local infrastructures allowed for during our campaign, the plume would be less diluted and $\Delta CO_2$ would show greater

enhancements, allowing for better relative precision than reported here.

Overall, our approach allows for detecting volcanic $CO_2$ enhancements with good confidence and for measuring volcanic HF, HCl, $SO_2$, BrO with high precision using a setup that can be easily deployed in the field in safe distance from the craters. Therefore, further refinements such as discussed above could make our approach a valuable tool in volcano monitoring. Inevitable drawbacks of direct-sun spectroscopy, though, are the required daytime and clear-sky conditions.

*Author contributions.* NB and AB developed the research question. AB, ASD, NB, JK, LF, CF, GBG, JK, PL, and LT took an active part in the field campaign by operating instrumentation, collecting, analyzing, and sharing data. FH and FK supported preparations for the field campaign through instrument developments. AB, ASD, NB, FH, FK, and QT contributed to the spectral retrievals. AB wrote the paper. ASD produced the figures for an earlier version of the manuscript.





*Acknowledgements.* We acknowledge support by the Heidelberg Karlsruhe Research Partnership (HEiKA) under project "Accurate Prototype remote sensing of correlated carbon and sulfur emissions from mount Etna (APE)" and Deutsche Forschungsgemeinschaft (DFG) under Emmy-Noether project "RemoteC" (BU2599/1-1). The data are available upon request from the authors.





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





**Table 1.** Spectral retrieval windows and their properties used for processing the FTS absorption spectra in the SWIR. Parentheses following absorber molecules indicate the used spectroscopic databases: TCCON refers to the spectroscopic database used by the Total Carbon Column Observing Network (Toon, 2014), HITRAN2008 refers to Rothman et al. (2009), HITRAN2009 refers to Rothman et al. (2009) with published updates and our own empirical modifications. LM refers to the line-mixing database by Lamouroux et al. (2010) used as a correction to HITRAN2008.

| Spectral range / $cm^{-1}$ | Target absorbers | Interfering absorbers |
|---|---|---|
| 7765–8005 | $O_2$ (TCCON), HF (HITRAN2008) | $H_2O$ (HITRAN2009), $O_2$-$O_2$ (TCCON) |
| 6308–6390 | $CO_2$ (LM) | $H_2O$ (HITRAN2009) $CH_4$ (HITRAN2008) |
| 6173–6276 | $CO_2$ (LM) | $H_2O$ (HITRAN2009) $CH_4$ (HITRAN2008) |
| 5684–5795 | HCl (HITRAN2008) | $CH_4$ (HITRAN2008) $H_2O$ (HITRAN2009) |

**Table 2.** Spectral retrieval windows and their properties used for processing the absorption spectra in the UV. Brackets following absorber molecules indicate the reference temperatures at which the absorption cross sections were measured in the laboratory. Parentheses contain references to the relevant data sources.

| Spectral range / nm | Target absorbers | Interfering absorbers |
|---|---|---|
| 312.0–326.8 | $SO_2$ [298 K] (Vandaele et al., 2009) | $O_3$ [221 K] (Burrows et al., 1999) |
| 330.6–352.8 | BrO [298 K] (Fleischmann et al., 2004) | $SO_2$ [298 K] (Vandaele et al., 2009) $O_3$ [221 K] (Burrows et al., 1999) $O_2$-$O_2$ (Hermans et al., 2003) $NO_2$ [294 K] (Vandaele et al., 1998) $CH_2O$ [298 K] (Meller and Moortgat, 2000) |



**Table 3.** Intra-plume VCD ratios found by fitting a straight line to the correlations shown in Figures 9 through 12. The errors are the standard deviations of the fitted slopes. The $R^2$ of the linear fit is shown in parentheses. On Sep. 22 and Sep. 23, we inferred VCD ratios for the whole day and 3 subsets separately. The $\Delta CO_2/SO_2$ ratios for the measurements after 7:45 UTC (bracketed) on Sep. 23 need to be considered with care since $\Delta CO_2$ is negative. For details see text.

| Date and time | $\Delta CO_2/SO_2$ | $SO_2/HF$ | $SO_2/HCl$ | $SO_2/BrO$ / $10^3$ |
|---|---|---|---|---|
| 2015-09-18 | 14.3±1.6 (0.45) | 10.5±0.3 (0.93) | 2.38±0.07 (0.94) | 5.54±0.12 (0.96) |
| 2015-09-22 | 6.1±2.4 (0.04) | 5.60±0.14 (0.91) | 1.74±0.04 (0.93) | 3.35±0.11 (0.86) |
| before 8:00 h UTC | 13.5±3.3 (0.20) | 6.53±0.22 (0.93) | 1.99±0.04 (0.97) | 4.68±0.12 (0.96) |
| between 8:00 and 8:30 h UTC | 7.1±1.5 (0.33) | 6.50±0.16 (0.97) | 2.01±0.05 (0.97) | 3.75±0.09 (0.97) |
| after 8:30 h UTC | 0.40±8.7 (0.00) | 4.76±0.13 (0.97) | 1.54±0.05 (0.96) | 3.26±0.18 (0.90) |
| 2015-09-23 | 35.2±2.3 (0.48) | 7.33±0.66 (0.32) | 2.52±0.09 (0.75) | 2.93±0.30 (0.28) |
| sunrise | 35.4±1.3 (0.88) | 21.2±0.3 (0.98) | 3.43±0.03 (0.99) | 12.5±0.4 (0.92) |
| between 7:00 and 7:45 h UTC | 11.6±2.8 (0.17) | 10.9±0.4 (0.89) | 2.42±0.05 (0.96) | 4.84±0.16 (0.92) |
| after 7:45 h UTC | [39.2±6.9 (0.34)] | 5.02±0.37 (0.74) | 1.54±0.08 (0.86) | 2.92±0.19 (0.78) |

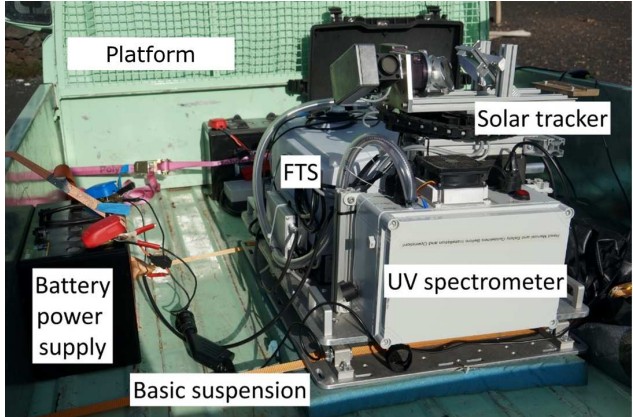

**Figure 1.** Photograph of the remote sensing observatory on the mobile platform, a small truck. The observatory consists of the Fourier Transform Spectrometer EM27/SUN and the UV spectrometer ULS2048x64-ENV5. A common solar tracker feeds sunlight into the two spectrometers. The observatory sits on a rubber and foam suspension. A 12-Volts battery supplies power to the spectrometers, the solar tracker, two control laptops for the spectrometers, and a low-power consumption PC for the solar tracker.





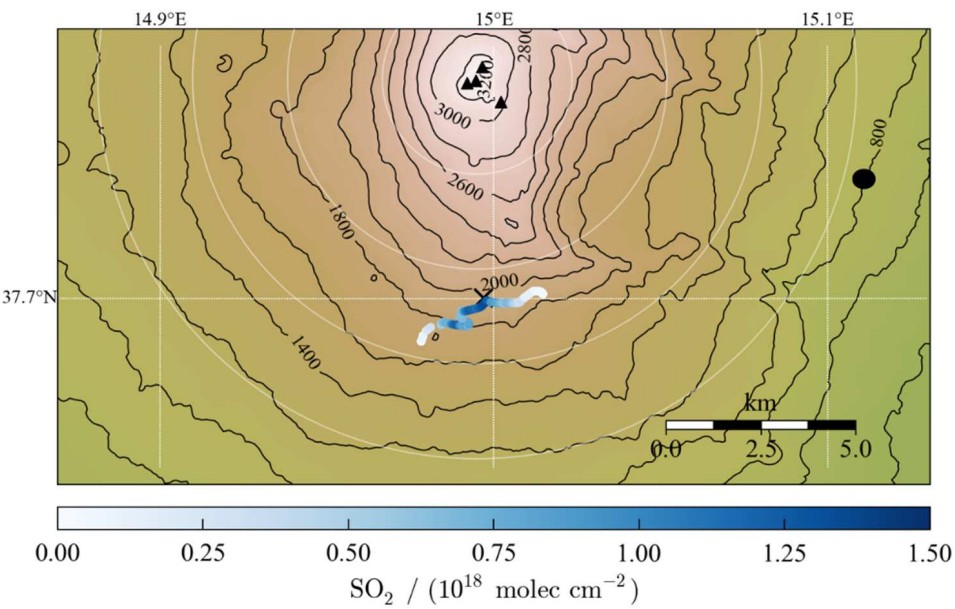

**Figure 2.** Topographic map of the southern slope of Mt. Etna. The background color shading corresponds to surface elevation with black contours in steps of 200 m altitude change. The white circular grid illustrates horizontal distances from the crater region in steps of 2.5 km as indicated by the ruler in the lower right. The 4 active craters on the summit at about 3,300 m altitude are indicated by black triangles and the base station at Milo is illustrated by the black circle on the eastern slope. The blueish trajectory in the middle of the map shows the motion of the mobile observatory on Sep. 22, between 8:00 and 8:50 h UTC (typical solar zenith angle SZA = 55°) in the vicinity of Rifugio G. Sapienza (black cross). The color code is a rough measure for the observed $SO_2$ VCDs as indicated by the colorbar. Note that $SO_2$ VCDs are plotted at the position of the observer at the time of measurement.





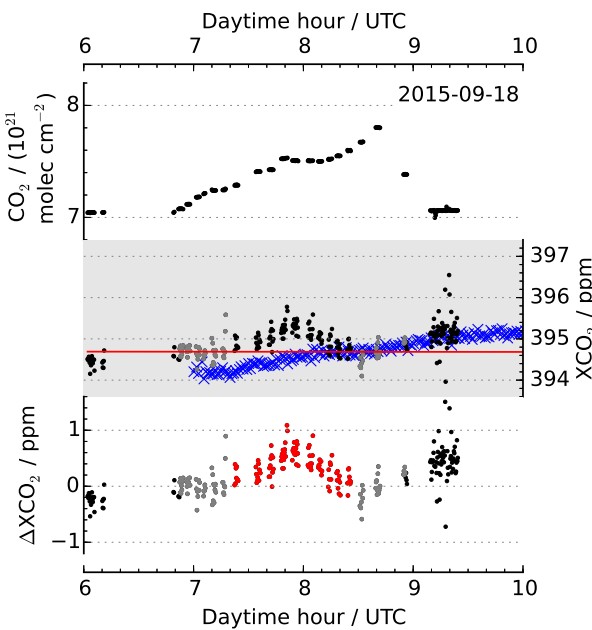

**Figure 3.** Derivation of $\Delta XCO_2$ for the spectra collected by the mobile FTS on Sep. 18, 2015. The upper panel shows the $CO_2$ VCDs retrieved by the spectral retrieval described in section 4.1. The middle panel shows the column-average dry-air mixing ratio $XCO_2$ (black and grey dots) calculated via equation (1). Background measurements (grey) are used to fit the linear background polynomial illustrated by the red line. Blue crosses show the measurements collected by the stationary FTS at the eastern slope of Mt. Etna. The lower panel depicts the volcanic $XCO_2$ enhancement ($\Delta XCO_2$) calculated according to equation (2). Red circles are considered intra-plume enhancements.





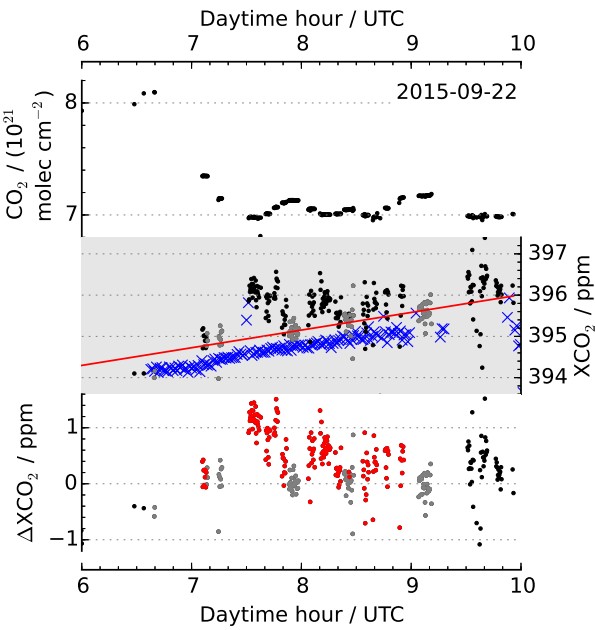

**Figure 4.** Same as Figure 3 for Sep. 22, 2015.

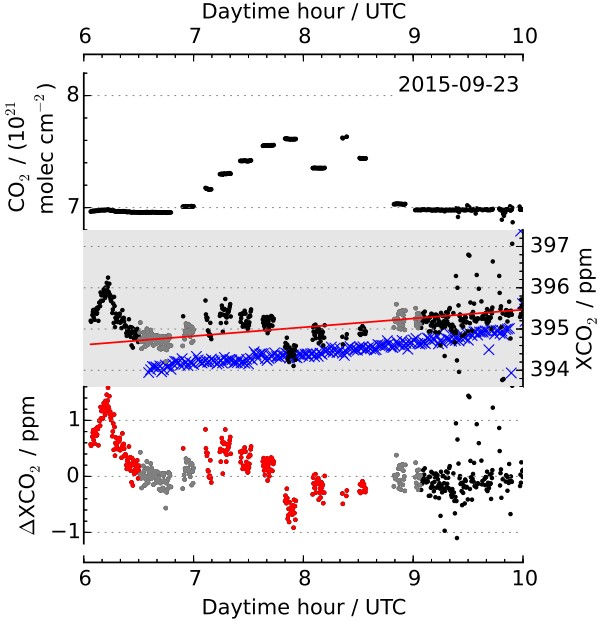

**Figure 5.** Same as Figure 3 for Sep. 23, 2015.



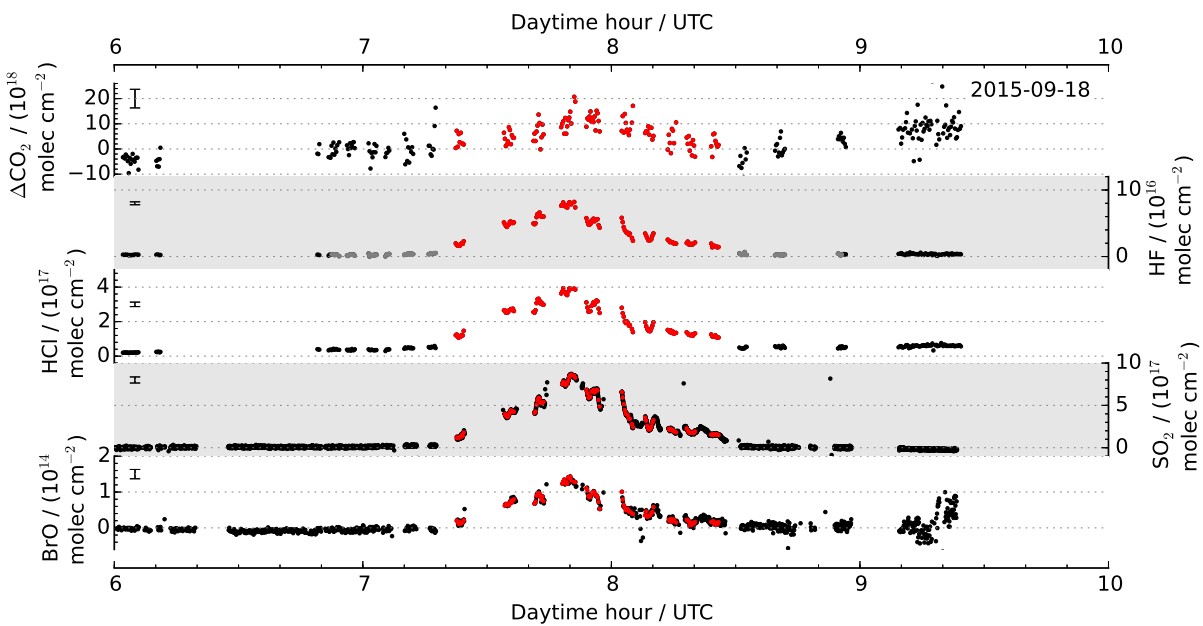

**Figure 6.** Time series of volcanic VCD enhancements in the plume of Mt. Etna observed on Sep. 18. $\Delta CO_2$ (first panel, left ordinate) correlates with enhancements in HF (second panel, right ordinate), HCl (third panel, left ordinate), $SO_2$ (fourth panel, right ordinate), and BrO (fifth subpanel, left ordinate). Grey symbols for HF indicate background measurements. Red closed symbols indicate intra-plume soundings where the UV-measured species $SO_2$ and BrO are averaged over the integration time of coincident FTS soundings. The precision estimated from the standard deviation of all background soundings is shown as an error bar in the upper left corner of each panel.





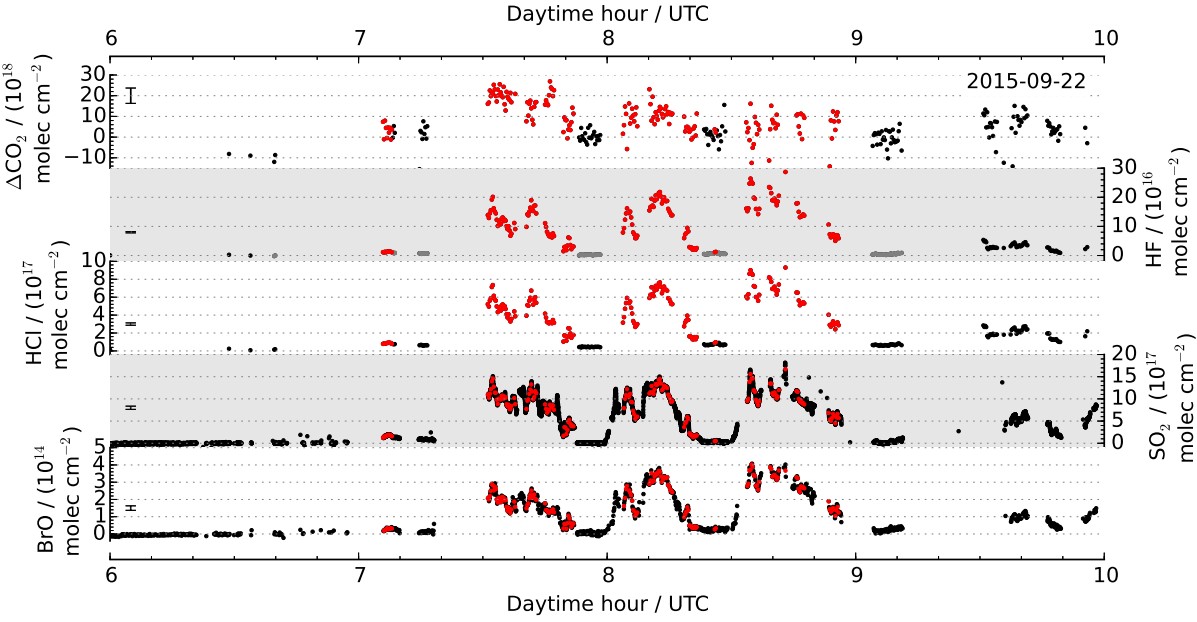

**Figure 7.** Same as Figure 6 for Sep. 22, 2016.

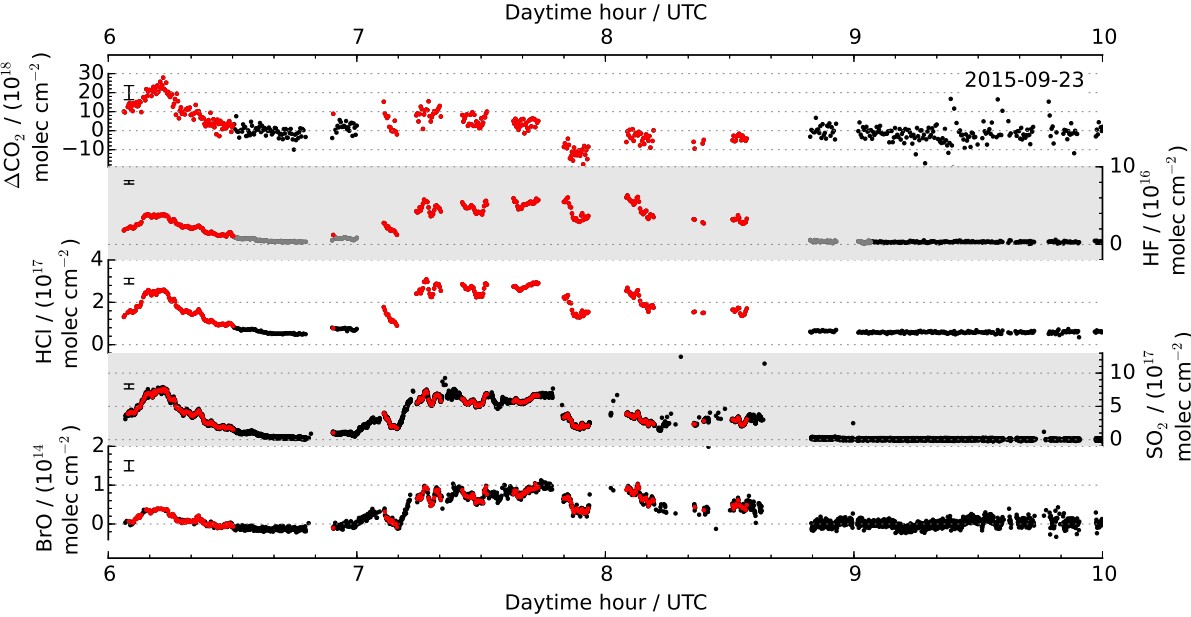

**Figure 8.** Same as Figure 6 for Sep. 23, 2016.





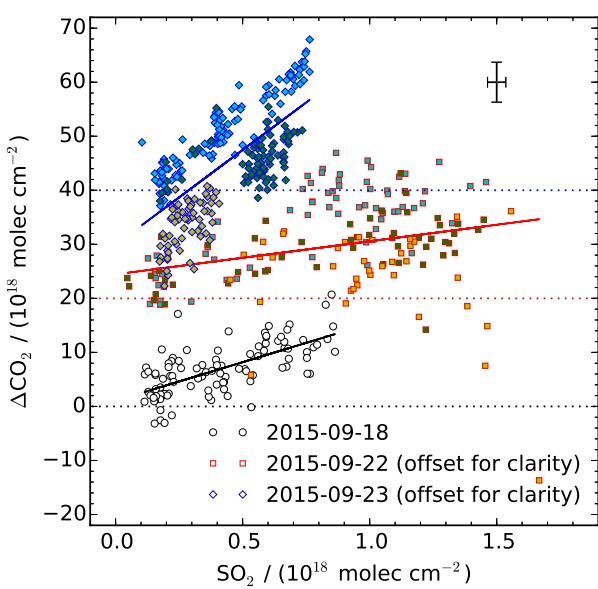

**Figure 9.** Correlation of $\Delta CO_2$ and $SO_2$ intra-plume VCDs observed on Sep. 18 (black circles), Sep. 22 (red squares), Sep. 23 (blue diamonds), 2015. On Sep. 22 and 23, we identify three separate groups. On Sep. 22, these groups are data recorded before 8:00 h UTC (cyan filling), between 8:00 and 8:30 h UTC (green filling), and after 8:30 h UTC (yellow filling). On Sep. 23, these groups are data recorded during sunrise (cyan filling), after sunrise before 7:45 h UTC (green filling), and after 7:45 h UTC (yellow filling). The solid lines are fits to the daily records, fits for the intra-day subsets are omitted for clarity. The error bars in the upper right corner illustrate the precision estimated from the standard deviation of background soundings. For clarity, data on Sep. 22 and Sep. 23 are offset in ordinate by $20 \times 10^{18}$ molec cm$^{-2}$ and $40 \times 10^{18}$ molec cm$^{-2}$, respectively, as indicated by the colored dotted lines.





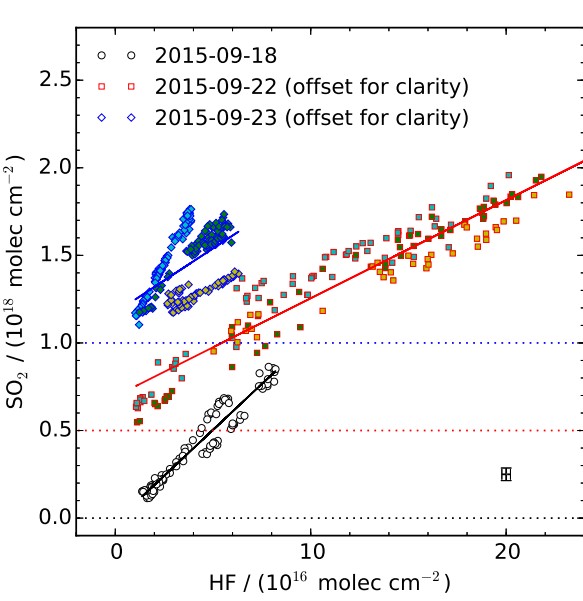

**Figure 10.** Same as Figure 9 but for $SO_2$ and HF VCDs. For clarity, data on Sep. 22 and Sep. 23 are offset in ordinate by $0.5 \times 10^{18}$ molec cm$^{-2}$ and $1.0 \times 10^{18}$ molec cm$^{-2}$, respectively, as indicated by the colored dotted lines.





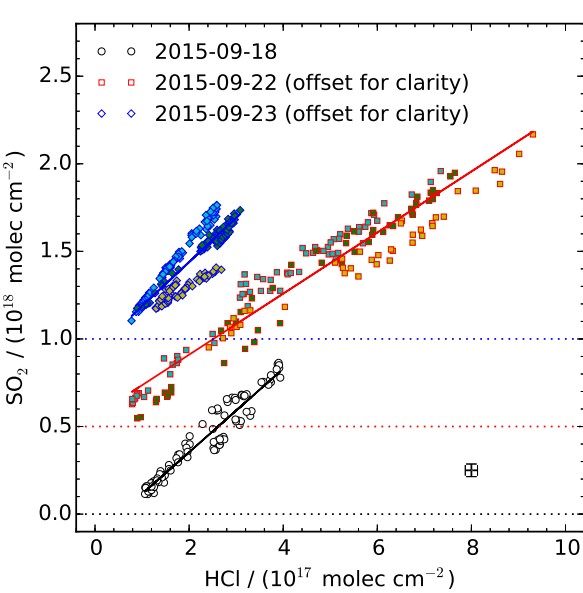

**Figure 11.** Same as Figure 9 but for $SO_2$ and HCl VCDs. For clarity, data on Sep. 22 and Sep. 23 are offset in ordinate by $0.5 \times 10^{18}$ molec cm$^{-2}$ and $1.0 \times 10^{18}$ molec cm$^{-2}$, respectively, as indicated by the colored dotted lines.





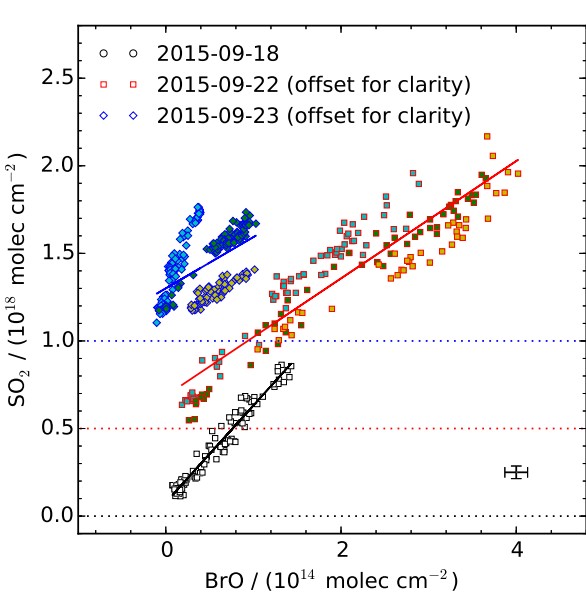

**Figure 12.** Same as Figure 9 but for $SO_2$ and BrO VCDs. For clarity, data on Sep. 22 and Sep. 23 are offset in ordinate by $0.5 \times 10^{18}$ molec cm$^{-2}$ and $1.0 \times 10^{18}$ molec cm$^{-2}$, respectively, as indicated by the colored dotted lines.