# Peer review of "Remote sensing of volcanic CO2, HF, HCl, SO2, and BrO in the downwind plume of Mt. Etna"

_Atmospheric Measurement Techniques, 2016_

## Short Comment (SC1) · 16 Sep 2016

General comments:

The paper presents results of a measurement survey, sensing volcanic gas concentrations (CO2, HF, HCl, SO2 and BrO) a couple of km downwind Mt. Etna, Sicily (Italy), using passive remote sensing apparatus. In particular, a fourier transform spectrometer operating in the SWIR, and a DOAS operating in the UV were employed.

I have not come across ground based passive remote sensing of volcanic gas from this long distance from the volcano mouth. While the methods used are established the data are new and an important result for climate related science and environmental monitoring in general. The paper is well written in a concise and straightforward manner.

I like the fact that the authors measured other volcanic species (HCL, HF etc.) in parallel as this allows a rather precise distinction of the volcanic CO2 in space.

The measurement precision, particularly that of CO2 VCD is impressive, given that the observatory was moving on a road (even though CO2 absorption spectra were recorded when the car stopped). How sensitive is the setup to shocks and vibrations? Is there an influence on the vibrations and have you quantified them or at least have a semi-quantitative measure?

The retrieval algorithm used remains mysterious, as well as the impact of model and fitting errors on the VCDs (see specific comments).

In my opinion the paper could be further improved by comparing the measured enhancements with plume dispersion models (e.g. Burton et al., 2013, p. 325), which are in line with your result. But this might be out of scope for an AMT paper.

Specific comments:

P2, I 22 and 23: Lidar BILLY measures range resolved CO2 concentrations. Whether or not it has to be close to the source depends on various parameters, including instrumental parameters such as the pulse energy, excess CO2 concentration, aerosol density etc.. There is no fundamental reason why the LIDAR has to be close to the plume. The CO2 plume of big cities is visible in airborne LIDAR signals from several km flight altitude.

P4, I31: How sensitive is the measurement precision of the gases, particularly CO2 (i.e. the  $3.7 \times 1018$  molec/cm2) to errors of your atmospheric model. Do the assumptions of your model (e.g. horizontally homogeneous layers) cause a bias? Are you actually able to quantify that bias since you do not know the "true" atmospheric composition (e.g. transmittance at a given wavelength etc.).

P5, I9: I do not understand the phrase, seems like the subject is missing

P5, I27: It would be interesting to know how high the fitting error is and how and if it
propagates into VCD.

P6, I8: It is clear that the measured VCD vary with observer altitude, but it is not very clear why this is a challenge to obtain the volcanic enhancement. Isn't the VCD looking through the plume larger than when looking outside of the plume (at constant observer altitude)?

P7: Section 5 is largely a discussion rather than pure result section and as such it might be better placed in the discussion section.

P9, I17: How did you estimate the measurement precision? P7, I29 does not really make it clear.

P9, I23: "The O 2 column was used to compensate CO 2 variations due to changes in observer altitude." What means "CO2 variations"? Variation in VCD?

P10, I10: Have you thought about measuring closer to the crater of Mt. Etna? Being  $\sim$ 1 km away the enhancement would be greater. Being off-roads, you would not be constrained to roads. This might allow assessing some of your sources of uncertainty (negative enhancements, minimum integration time etc.).

Technical comments:

P4, l8: direct or directed?

P13, I17, space before comma: FTIR measurements , J.

---

## Referee Comment (RC1) · Anonymous Referee #1 · 21 Oct 2016

This is a well written paper about the measurement of the volcanic gas composition of Mt Etna's with mobile remote sensing instrumentation. Although the concepts are not new since similar investigations have already been performed around the world, the authors do an excellent job describing the experimental and analytical techniques employed. Furthermore, the advances made recently in the improvement of commercially available instrumentation (i.e. robust spectrometers, refined solar tracking systems) was taken advantage of in order to get unprecedented precision in the detection of gases like CO2 from the volcanic plume via the solar absorption technique. Much is still to be improved, as the authors comment, but this manuscript provides a very good insight of what is possible in terms of quality of the results with the now available technology. The article should be published in AMT after the following recommendations are taken into consideration.

- Give more detail on the optical set-up of the UV spectrometer/telescope in p4 l4 and provide a reference if available.

- Nothing is mentioned about the plume heights during the days of the experiments. It would be useful to know for future studies considering that the sensitivity of the technique is strongly dependent on the true distance to the plume and thus the dilution effect.

- p5 I7. Why not use the background VCD-scaled profiles from the stationary FTS measurements instead of the a priori profiles?

- p5 l9. By "lower tropospheric" part you mean one or several layers in your RT model? Please specify how the constraint is set.

- p6 I10. What about atmospheric pressure. Does small variations in the detected VCD's also vary with surface pressure?

- Figs4&5. There are some intra-plume dXCO2 values which fall in negative values with as much as 1 ppm, while the precision is reported to be considerably smaller (3.7x1018 molec/cm2). This value should also be converted to delta XCO2 (ppm) to have an idea. The authors argument later that the transition from CO2-rich to CO2-poor plume might be the reason for negative values, while the other gases remain elevated. But if HF is being used as proxy for intra-plume measurements, then this doesn't explain it. Maybe a dCO2/SO2 and dCO2/HF time series for these critical phases could support the idea that strong plume composition variations are happening within the same morning.

- p7 I14. The authors state that "An improved setup might benefit from operating the stationary FTS closer to the trajectory of the mobile observatory". The way it is explained, the grey points on the plots from the mobile instruments are used to produce the time-dependent background function (P). So, it is not clear to me what is meant by this. Is it the difference between the red line and the blue crosses that they want to

minimize? What for if it is not used in the calculations?

- p10 l8. I don't see how wind direction could affect the observed variations. Should't the plume contain a mixture of all craters 5-10 km downwind?

- Figs6-8. There are clear intra-plume measurements of SO2 and BrO which are not red-colored. Please include an explanation in the text why this is so, I couldn't find it.

- Fig9. the offset in the ordinate for the different days should be made clearer in the plot and not only in the caption. Maybe they can start labelling the y-axis at 0 each time where the dotted lines are drawn.

- It would be useful for certain readers, maybe geochemists or other researchers who want to compare their previos investigations, to include average ratios for the entire campaign in the bottom of table 3.

- At the end it was unclear to me the benefit of having a fast-scan solar tracker if only stationary data during the stop-and-go periods were utilised for calculating the ratios. Minor

- The paragraph in section 2 is too long, break into at least two paragraphs.

- p3 l32 . . . in order to increase measurement frequency.

- p4 I10. Compared to applications which require stationary deployment, volcanic ...

- p4 l26. ...due to unstable tracking of the sun (or clouds passing by ?!)

- p4 l32 along the vertical, -> along the optical path, and derived the corresponding vertical-columnn-densities (VCDs).

- p5 I19. Climatological profiles from a specific model? Please specify.

- p5 l29. were dark spectra measured and subtracted?

- p5 l30. remove "such"

---

## Author Comment (AC1) · 24 Oct 2016

Manuscript: "Remote sensing of volcanic $CO_2$, HF, HCl, $SO_2$, and BrO in the downwind plume of Mt. Etna" by A. Butz et al.

Reply to interactive comment by M. Queisser

We thank Dr. Queisser for his comments on our manuscript. Please find our point-by-point reply below. We will consider modifications of the manuscript after all reviews will have become available.

General comments:

<<*The paper presents results of a measurement survey, sensing volcanic gas concentrations ($CO_2$, HF, HCl, $SO_2$ and BrO) a couple of km downwind Mt. Etna, Sicily (Italy), using passive remote sensing apparatus. In particular, a fourier transform spectrometer operating in the SWIR, and a DOAS operating in the UV were employed. I have not come across ground based passive remote sensing of volcanic gas from this long distance from the volcano mouth. While the methods used are established the data are new and an important result for climate related science and environmental monitoring in general. The paper is well written in a concise and straightforward manner.*

*I like the fact that the authors measured other volcanic species (HCl, HF etc.) in parallel as this allows a rather precise distinction of the volcanic $CO_2$ in space. The measurement precision, particularly that of $CO_2$ VCD is impressive, given that the observatory was moving on a road (even though $CO_2$ absorption spectra were recorded when the car stopped). How sensitive is the setup to shocks and vibrations? Is there an influence on the vibrations and have you quantified them or at least have a semi-quantitative measure?*>>

The reason why we had to stop the truck for $CO_2$ measurements was that shocks and vibrations prevented the solar tracker from reliably tracking the sun over the 12 second integration time of the FTS. Heavy vibrations and shocks caused entire loss of the solar tracking. Lighter vibrations that caused brightness fluctuations could be monitored via the DC-part of the recorded interferograms. The later effect was used to discard such data.

The EM27/SUN FTS itself proved remarkably resistant against shocks and vibrations. It did not require any optical realignment during the entire campaign deployment. We monitored optical alignment through measurements of the instrument line shape (ILS) along a procedure for example described in Klappenbach et al., 2016. To this end, we operated an ordinary halogen lamp in a few meters distance from the spectrometer and coupled the light into the spectrometer via the solar tracker mirrors. Then, we fitted a model of the instrument line shape to water vapor absorption lines around wavenumber 7000-7400 cm$^{-1}$ (1.4 micron wavelength) . The ILS did not show any significant changes during the campaign.

Thus, there is no direct influence of shocks and vibrations on the reported $CO_2$ records but recording was only possible when the truck was stopped.

<<*The retrieval algorithm used remains mysterious, as well as the impact of model and fitting errors on the VCDs (see specific comments).*>>

The retrieval algorithms PROFFIT and DOAS are standard methods in the atmospheric measurement community and our paper provides the relevant references. We would prefer not to expand on these technical aspects further.

Concerning the error estimates, we took the standard deviation of all background soundings as a measure for the retrieval error. This is conservative with respect to fitting errors, since our estimates thereby include instruments noise, residual fluctuations of solar tracking quality, as well as residual variability of the background gas. Systematic errors such as discussed for the determination of the $CO_2$ enhancement (removal of surface elevation effect and gas background) are not part of the estimated retrieval error. Our paper discusses these assumptions and caveats.

<<*In my opinion the paper could be further improved by comparing the measured enhancements with plume dispersion models (e.g. Burton et al., 2013, p. 325), which are in line with your result. But this might be out of scope for an AMT paper.*>>

We agree. But indeed, we find it out of scope for the present paper to run a plume dispersion model. Our goal here is to demonstrate the capabilities of our measurement technique in terms of detection accuracy for volcanic $CO_2$ and with respect to the number of volcanic gases that can be observed simultaneously.

Since we believe that the technique is of great interest for others working in the field of volcano remote sensing, we chose for a relatively quick and direct path to publication.

Specific comments:

**<<P2, l 22 and 23: Lidar BILLY measures range resolved $CO_2$ concentrations. Whether or not it has to be close to the source depends on various parameters, including instrumental parameters such as the pulse energy, excess $CO_2$ concentration, aerosol density etc.. There is no fundamental reason why the LIDAR has to be close to the plume. The $CO_2$ plume of big cities is visible in airborne LIDAR signals from several km flight altitude.>>**

We will change the manuscript taking this comment into account. In the manuscript, though, it says "Thus, they need to sample the plume in the proximity of the source." Our intention was to refer to the fact that techniques such as a LIDAR need to point toward the source and lines-of-sight must cross the plume close to source where path-integrated $CO_2$ enhancements are still much larger than several ppm. So, we did not directly refer to the location of the LIDAR which indeed can be deployed far away if the local environment allows for unobstructed view to the source and if a suitable reflection target is available. But, we understand that our statement might lead to misunderstanding.

**<<P4, l31: How sensitive is the measurement precision of the gases, particularly $CO_2$ (i.e. the $3.7\times10^{18}$ molec/cm2) to errors of your atmospheric model. Do the assumptions of your model (e.g. horizontally homogeneous layers) cause a bias? Are you actually able to quantify that bias since you do not know the "true" atmospheric composition (e.g. transmittance at a given wavelength etc.).>>**

The measurement precision (i.e. the random error component) is not affected by our retrieval assumptions.

However, our assumptions on plume geometry and atmospheric layering can introduce a systematic error if the reported VCDs were used to calculate the volcanic emission flux without considering plume geometry. The reported VCDs amount to the "true" volcanic enhancements if the assumption holds that the lines-of-sight cross the plume perpendicularly to its propagation direction and that the plume is homogeneous (page 6, line 1 ff). For inferring emission fluxes, one would need to run a plume model which provides plume geometry, propagation direction, and gas inhomogeneity. This could be used as "true" atmosphere in the retrieval or for translating the retrieved VCDs into a (slant) path-integrated enhancement. Since we argue that running a plume model is out-of-scope for this study and since it comes with its own uncertainties, we chose to report VCDs as defined in the paper. This approach should be sufficiently clear to enable follow-up studies.

We want to emphasize (as in the paper page 6, line 4), that ratios of gases are not affected by these assumptions if the gases share the same distribution in the plume.

**<<P5, l9: I do not understand the phrase, seems like the subject is missing>>**

"For $CO_2$, HCl, and HF, we only scaled the lower tropospheric part of the vertical profile and adopted the a priori for the upper tropospheric and stratospheric part."
->
For $CO_2$, HCl, and HF, we only scaled the lower tropospheric part of the vertical profile while the upper tropospheric and stratospheric part was taken from the a priori."

**<<P5, l27: It would be interesting to know how high the fitting error is and how and if it propagates into VCD.>>**

As outlined above, our error estimate is empirically derived and includes various error sources. In particular, the noise error calculated by PROFFIT and the fitting error calculated by DOAS are smaller than our top-down error estimate.

**<<P6, l8: It is clear that the measured VCD vary with observer altitude, but it is not very clear why this is a challenge to obtain the volcanic enhancement. Isn't the VCD looking through the plume larger than when looking outside of the plume (at constant observer altitude)?>>**

Indeed, at constant observer altitude, no $O_2$ correction was necessary to calculate $CO_2$ enhancements. But, intra-plume and background soundings were never taken at the same observer altitude. Mt. Etna's rough

topography implies that lateral displacements of the observing platform virtually always go along with changes in observer altitude which cannot be neglected.

**<<P7: Section 5 is largely a discussion rather than pure result section and as such it might be better placed in the discussion section.>>**

We tried to separate the discussion of quantitative results (section 5) from more qualitative statements on our work and how it links to past and future efforts (section 6). We believe this separation supports a clear structure of the manuscript.

**<<P9, l17: How did you estimate the measurement precision? P7, l29 does not really make it clear.>>**

Page 7, Line 29: "We estimated the overall $\Delta CO_2$ precision from the standard deviation (1-$\sigma$) of all the background measurements identified via the HF threshold."

We divided our data record in background soundings and intra-plume soundings. The background measurements are those which show negligible HF concentrations assuming that detection of HF is a good indicator for our lines-of-sight crossing the volcanic plume. The intra-plume measurements are those with significant HF amounts. As described in section 4.3, in particular equation (2), we used the background soundings to subtract the $XCO_2$ background concentration and to calculate the $CO_2$ enhancement $\Delta XCO_2$.

For estimating the precision, we simple calculated the standard deviation of all background soundings. We will consider adding some more text in the revised manuscript.

**<<P9, l23: "The $O_2$ column was used to compensate $CO_2$ variations due to changes in observer altitude." What means "$CO_2$ variations"? Variation in VCD?>>**

Yes, "Variation in VCD" is correct.

**<<P10, l10: Have you thought about measuring closer to the crater of Mt. Etna? Being _1 km away the enhancement would be greater. Being off-roads, you would not be constrained to roads. This might allow assessing some of your sources of uncertainty (negative enhancements, minimum integration time etc.).>>**

Yes, we have considered this option for future work (page 10, line 23) and, indeed, being somewhat closer to the source would enhance our signal-to-noise ratio. But, of course, we aim at remote sensing in safe distance from the crater with small logistics overhead. Going off-road closer to the crater counteracts these goals. But, thinking of a permanent, stationary remote sensing network, distances of ~3 km instead of 5-10 km might be better suited.

**<<Technical comments:**
**P4, l8: direct or directed?>>**

"direct" since it is a general explanation how the setup works, not only for our study but for essentially all EM27/SUN work.

**<<P13, l17, space before comma: FTIR measurements , J.>>**

Yes.

---

## Author Comment (AC2)

Manuscript: "Remote sensing of volcanic $CO_2$, HF, HCl, $SO_2$, and BrO in the downwind plume of Mt. Etna" by A. Butz et al.

Reply to comment by anonymous referee #1 (all page and line number refer to the AMTD version of the manuscript).

We thank the referee for the comments on our manuscript. Please find our point-by-point reply below.

*<<This is a well written paper about the measurement of the volcanic gas composition of Mt Etna's with mobile remote sensing instrumentation. Although the concepts are not new since similar investigations have already been performed around the world, the authors do an excellent job describing the experimental and analytical techniques employed. Furthermore, the advances made recently in the improvement of commercially available instrumentation (i.e. robust spectrometers, refined solar tracking systems) was taken advantage of in order to get unprecedented precision in the detection of gases like $CO_2$ from the volcanic plume via the solar absorption technique. Much is still to be improved, as the authors comment, but this manuscript provides a very good insight of what is possible in terms of quality of the results with the now available technology. The article should be published in AMT after the following recommendations are taken into consideration.>>*

*<<- Give more detail on the optical set-up of the UV spectrometer/telescope in p4 l4 and provide a reference if available.>>*

We add some text on the UV spectrometer:

*Page 4:*
*The UV grating spectrometer is an AvaSpec-ULS2048x64-ENV5 manufactured by Avantes. It features a 1,800 lines/mm grating and 2048 pixel linear array CCD detector covering the spectral range 294 to 457 nm. The entrance slit of the spectrometer is 200 $\mu m$ wide supporting an optical resolution of 0.8 nm (FWHM). The spectrometer is placed inside a housekeeping box (EnviMesS TSE 1.1) equipped with a Peltier cooling that keeps the spectrometer temperature stable at 15.00°C with a precision of 0.02°C. The housekeeping box disposes of a glass fiber coupling that allows for externally connecting a glass fiber for light intake without the need to open the box. During our campaign, the exposure times for individual spectra under clear-sky were around 15 ms. Typically, we coadded 30 spectra leading to effective exposures of several hundred ms.*

We add some text and a figure providing information on the telescope:

[Figure]

*New Figure 3. Sketch of the telescope used to feed sunlight into the UV spectrometer. The telescope was mounted into the outer part of solar light beam collected by the solar tracker. A prism feeds sunlight into the telescope (0.5-inch tube diameter, ~120 mm tube length) which consists of a UV-filter (Hoya U330), a lens (focal length f=40 mm), and a circular aperture (800 $\mu m$ diameter) aligned to the focus of the lens. A polytetrafluoroethylene (PTFE) diffuser plate illuminates the glass fiber (400 $\mu m$ diameter) which then guides the sunlight into the UV spectrometer.*

*Page 4:*
*Figure 3 shows a sketch of the telescope assembly. A prism (10 mm sidelength) reflects the incoming sunlight into 0.5-inch lens tube mounted on a kinematic platform (not shown). The light beam first passes a UV-filter (Hoya U330) that shields unused parts of the solar spectrum, then a lens (focal length f=40 mm) focuses the beam on a 800 $\mu m$ wide, circular aperture. Further downstream, a polytetrafluoroethylene (PTFE) diffuser plate illuminates a 400 $\mu m$ glass fiber that takes the light into the spectrometer box via the fiber coupling. To support optical alignment, the length of the lens tube is adjustable, the prism can be rotated, and the attitude of the whole telescope can be controlled through the kinematic mount.*

*<<- Nothing is mentioned about the plume heights during the days of the experiments. It would be useful to know for future studies considering that the sensitivity of the technique is strongly dependent on the true distance to the plume and thus the dilution effect.>>*

Our technique delivers measurements of the total gas column between the observer and the sun. Thus, dilution in the vertical dimension should not affect the sensitivity of our technique (as long as the sun is roughly overhead) since the technique would still catch the volcanic surplus of molecules.

Dilution in the horizontal dimensions, however, does affect our sensitivity. We make note of this effect in the discussion and conclusion section where we argue that deploying the spectrometers closer to source would enhance our precision (page 10, line 24). Horizontal dilution depends on the prevailing wind patterns and thus, horizontal dilution also depends on the height of the plume. For the days reported here, the general observation was that there was no major buoyant rise of the plume out of the crater (as seen by visual inspection of the plume condensate) but that the plume remained roughly at the altitude level of the crater region (above 3,100 m) and that it dispersed horizontally. Our direct-sun observations provide some limited information on plume altitude which originates from observing the plume under various viewing angles. Preliminary analyses [Dinger, 2016] show that the plume was indeed located between 3,000 and 4,000 m altitude. While these investigations confirm our visual inspection, they are preliminary and thus, we would prefer not to include it in the manuscript.

*<<- p5 l7. Why not use the background VCD-scaled profiles from the stationary FTS measurements instead of the a priori profiles?>>*

The retrieval of scaling factors does not impose side-constraints i.e. the scaling factors for the gas profiles are determined in a least-squares sense. Therefore, using a different initial guess or a priori for these scaling factors does not affect the retrieval except for the number of required iterations. The latter could be minimized by choosing a better initial guess. Since our a priori scaling factors are reasonably close to the true ones, the effect on the number of iterations is negligible.

*<<- p5 l9. By "lower tropospheric" part you mean one or several layers in your RT model? Please specify how the constraint is set.>>*

We scaled 4 layers in the RT model between 3.2 and 4.9 km altitude. In general, we did multiple test runs including the use of a different retrieval algorithm, allowing for scaling larger parts of the tropospheric profile or even the entire profile. Differences between the test runs were minor compared to the detected volcanic signals.

> *Page 5, line 9:*
> *we only scaled the lower tropospheric part of the vertical profile and adopted the a priori for the upper tropospheric and stratospheric part.*
> *->*
> *we only scaled the lower tropospheric part of the vertical profile (four layers between 3.2 and 4.9 km altitude) and adopted the a priori for the rest*

*<<- p6 l10. What about atmospheric pressure. Does small variations in the detected VCD's also vary with surface pressure?>>*

Yes, variations in atmospheric pressure would have a similar effect as variations of observer altitude. Ratioing by the retrieved $O_2$ column (equation (1)) takes this effect into account as well. For our setup, however, the topographic variations are large compared to meteorological effects.

We change the manuscript

> *Page 6, line 10:*
> *co-vary with observer altitude*
> *->*
> *co-vary with surface pressure, for our setup mainly variable due to variable observer altitude,*

*<<- Figs4&5. There are some intra-plume dXCO$_2$ values which fall in negative values with as much as 1 ppm, while the precision is reported to be considerably smaller ($3.7x10^{18}$ molec/cm$^2$). This value should also be converted to delta XCO$_2$ (ppm) to have an idea.>>*

We add the precision estimate in units of ppm:

> *Page 7, line 30:*

*It amounts to 3.7×10$^{18}$ molec/cm$^2$ suggesting that individual ΔCO$_2$ measurements exceed the precision estimate by a factor 5 to 6.*
*->*
*It amounts to 3.7×10$^{18}$ molec/cm$^2$ for ΔCO$_2$ (0.20 ppm for ΔXCO$_2$) suggesting that individual measurements exceed the precision estimate by a factor 5 to 6.*

Please note that we use the word "precision" referring to the random error component. Negative values such as in Figs. 4 and 5 are most likely due to systematic effects as suggested in the manuscript (page 8, line 32).

**<<The authors argument later that the transition from CO$_2$-rich to CO$_2$- poor plume might be the reason for negative values, while the other gases remain elevated. But if HF is being used as proxy for intra-plume measurements, then this doesn't explain it. Maybe a dCO$_2$/SO$_2$ and dCO$_2$/HF time series for these critical phases could support the idea that strong plume composition variations are happening within the same morning.>>**

Figs. 9 through 12 include information on the measurement time through the colored filling of the symbols. On Sep. 23, Fig. 9 shows that the early transect (blue symbols, green filling) yields relatively high CO$_2$ and high SO$_2$. The later transect (blue symbols, yellow filling) shows low CO$_2$ (even negative) and low SO$_2$. Likewise, Fig. 10 shows relatively high SO$_2$ and high HF for the early transect (blue symbols, green filling). For the later transect (blue symbols, yellow filling), however, HF remains high while SO$_2$ is low. Thus, we would argue that Figs. 9 through 12 support our statement on changing plume composition.

**<<- p7 l14. The authors state that "An improved setup might benefit from operating the stationary FTS closer to the trajectory of the mobile observatory". The way it is explained, the grey points on the plots from the mobile instruments are used to produce the time-dependent background function (P). So, it is not clear to me what is meant by this. Is it the difference between the red line and the blue crosses that they want to minimize? What for if it is not used in the calculations?>>**

Indeed, the measurements of the stationary FTS (blue crosses) are not used for any calculations. We included them in the plots to justify our assumption on a smoothly varying background concentration (fitted by a straight line to the "grey" background measurements of the mobile FTS).

Initially, we planned to use the stationary FTS measurements directly for background removal. Using the independent background measurements would have the advantage that we would not need to interpolate between background measurement before and after a plume transect. However, we refrained from doing so, since there is a small (so far unexplained) difference between the stationary FTS and the mobile background measurements. In the manuscript, we speculate that this difference might be due to the stationary and the mobile FTS observing different (background) air composition. Bringing the two instruments closer to each other would reduce this source of uncertainty.

**<<- p10 l8. I don't see how wind direction could affect the observed variations. Should't the plume contain a mixture of all craters 5-10 km downwind?>>**

Mt. Etna's crater region consists of four larger craters that are separated by several hundred meters in the horizontal (up to a kilometer between the North-East and the South-East crater) which is smaller than our sampling distance but not negligible. Further, the craters are somewhat displaced in the vertical and thus, they might be subject to different wind patterns. Voigt et al., 2014, also observed incomplete mixing of plumes at Mt. Etna (see in particular Fig. 1 in Voigt et al., 2014). Thus, wind direction can matter for plume observations at Mt. Etna. We add the reference to Voigt et al., 2014, at page 10, line 4.

**<<- Figs6-8. There are clear intra-plume measurements of SO$_2$ and BrO which are not red-colored. Please include an explanation in the text why this is so, I couldn't find it.>>**

We add the information in caption of Figure 6.

*Fig. 6, caption:*
*Red closed symbols indicate intra-plume soundings where the UV-measured species SO$_2$ and BrO are averaged over the integration time of coincident FTS soundings.*
*->*
*Red closed symbols indicate intra-plume soundings used for further interpretation. For the UV-measured species SO$_2$ and BrO, we only consider those intra-plume soundings that occur within the integration time of a coincident FTS measurement. Therefore, some SO$_2$ and BrO measurements that are clearly intra-plume are not red colored.*

*<<- Fig9. the offset in the ordinate for the different days should be made clearer in the plot and not only in the caption. Maybe they can start labelling the y-axis at 0 each time where the dotted lines are drawn.>>*

The dotted lines in Figs. 9-12 indicate the offsets graphically (in addition to the text in the caption). In our opinion, putting multiple 0 labels on the y-axis is rather confusing.

*<<- It would be useful for certain readers, maybe geochemists or other researchers who want to compare their previos investigations, to include average ratios for the entire campaign in the bottom of table 3.>>*

We include the average VCD-ratios in table 3 (table and caption changed).

However, those averages have to be considered with caution. The VCD-ratios reported in table 3 are highly variable. The origin of variability is not measurement noise that averages out but rather geophysical variability of the parameters that volcanologists, geochemists and other scientists are interested in. Thus, taking campaign-average values as basis for further interpretation is inadequate.

*<<- At the end it was unclear to me the benefit of having a fast-scan solar tracker if only stationary data during the stop-and-go periods were utilised for calculating the ratios.>>*

The main benefit of the fast solar tracker was that it enabled $SO_2$ measurements while driving (page 4, line 13). Thereby, the operator on the passenger seat was able to tell whether the mobile spectrometers sampled intra- or extra-plume air masses. Thus, the fast-scan solar tracker was of great help for operating stop-and-go patterns. This aspect might be considered scientifically marginal, logistically and technically it was a key enabling aspect.

Of course, we hoped for being able to also measure $CO_2$ while driving. Although we did not succeed in this aspect, we gained valuable insight in how to improve the tracker for future campaigns.

*<<Minor*
*- The paragraph in section 2 is too long, break into at least two paragraphs.>>*

Done.

*<<- p3 l32 …in order to increase measurement frequency.>>*

Added.

*<<- p4 l10. Compared to applications which require stationary deployment, volcanic …>>*

*Compared to previous stationary and ship-borne applications …*

*<<- p4 l26. … due to unstable tracking of the sun (or clouds passing by ?!)>>*

*due to unstable tracking of the sun or thin clouds passing by.*

*<<- p4 l32 along the vertical, -> along the optical path, and derived the corresponding vertical-columnn-densities (VCDs).>>*

We kept the text as it was. The algorithm retrieves VCDs. The sentence suggested by the reviewer implies a two-step procedure which is not how it works.

*<<- p5 l19. Climatological profiles from a specific model? Please specify.>>*

We add

> *Page 5, line 19:*
> *we used climatological profiles*
> *->*
> *we used climatological profiles taken from the recommendations of the Infrared Working Group (IRWG) of the Network for the Detection of Atmospheric Composition Change (NDACC) for the mid-latitudes.*

*<<- p5 l29. were dark spectra measured and subtracted?>>*

Yes. We add

> *… after subtraction of dark current and offset spectra.*

**<<- p5 l30. remove "such">>**

Done.

**Additional References:**

Dinger S., Direct sunlight spectroscopy of volcanic plume composition in the downwind plume of Mt. Etna, Master thesis, Heidelberg University, Germany, 2016

Voigt, C., P. Jessberger, T. Jurkat, S. Kaufmann, R. Baumann, H. Schlager, N. Bobrowski, G. Giuffrida, and G. Salerno, Evolution of $CO_2$, $SO_2$, HCl, and $HNO_3$ in the volcanic plumes from Etna, Geophys. Res. Lett., 41, 2196–2203, doi:10.1002/2013GL058974, 2014